

# The multi metal-resistant bacterium *Cupriavidus metallidurans* CH34 affects growth and metal mobilization in *Arabidopsis thaliana* plants exposed to copper

Claudia Clavero-León[1,2], Daniela Ruiz[1,2], Javier Cillero[1,2], Julieta Orlando[3] and Bernardo González[1,2]

[1] Laboratorio de Bioingeniería, Facultad de Ingeniería y Ciencias, Universidad Adolfo Ibáñez, Santiago, Chile
[2] (CAPES), Center of Applied Ecology and Sustainability, Santiago, Chile
[3] Laboratorio de Ecología Microbiana, Departamento de Ciencias Ecológicas, Facultad de Ciencias, Universidad de Chile, Santiago, Chile

## ABSTRACT

Copper (Cu) is important for plant growth, but high concentrations can lead to detrimental effects such as primary root length inhibition, vegetative tissue chlorosis, and even plant death. The interaction between plant-soil microbiota and roots can potentially affect metal mobility and availability, and, therefore, overall plant metal concentration. *Cupriavidus metallidurans* CH34 is a multi metal-resistant bacterial model that alters metal mobility and bioavailability through ion pumping, metal complexation, and reduction processes. The interactions between strain CH34 and plants may affect the growth, metal uptake, and translocation of *Arabidopsis thaliana* plants that are exposed to or not exposed to Cu. In this study, we looked also at the specific gene expression changes in *C. metallidurans* when co-cultured with Cu-exposed *A. thaliana*. We found that *A. thaliana*'s rosette area, primary and secondary root growth, and dry weight were affected by strain CH34, and that beneficial or detrimental effects depended on Cu concentration. An increase in some plant growth parameters was observed at copper concentrations lower than 50 μM and significant detrimental effects were found at concentrations higher than 50 μM Cu. We also observed up to a 90% increase and 60% decrease in metal accumulation and mobilization in inoculated *A. thaliana*. In turn, copper-stressed *A. thaliana* altered *C. metallidurans* colonization, and *cop* genes that encoded copper resistance in strain CH34 were induced by the combination of *A. thaliana* and Cu. These results reveal the complexity of the plant-bacteria-metal triad and will contribute to our understanding of their applications in plant growth promotion, protection, and phytoremediation strategies.

Corresponding author
Bernardo González,
bernardo.gonzalez@uai.cl

## INTRODUCTION

Copper (Cu), like many other elements, is an essential trace nutrient required for normal plant growth and development (*Silver, 1998*; *Andresen, Peiter & Küpfer, 2018*; *Shabbir et al., 2020*; *Kumar et al., 2021*). Required Cu levels in plants range between 5 and 30 mg $kg^{-1}$, whereas Cu levels in agricultural soils normally range between 10 and 60 mg $kg^{-1}$ (*Ballabio et al., 2018*). Unnatural soil copper sources include manure, sewage sludge, and mineral fertilizers and pesticides (*Oorts, 2013*), and copper accumulation and leaching are strongly dependent on soil type and conditions (*Shabbir et al., 2020*). High copper levels negatively affect plants and microorganisms thriving in soil environments (*Rajput et al., 2020*; *Shabbir et al., 2020*), leading to the development and expression of detoxification and tolerance processes (*Giachino & Waldron, 2020*; *Kumar et al., 2021*).

Because of its redox potential, Cu has the ability to cycle between oxidized $Cu^{2+}$ and reduced $Cu^+$, thus mediating electron transfer reactions as a cofactor in the active centers of several key enzymes. $Cu^+$ reduction places Cu in crucial roles in environmental and physiological processes such as photosynthesis, respiration, reactive oxygen species detoxification, ethylene perception, and cell wall remodeling and lignification (*Maksymiec, 1998*; *Burkhead et al., 2009*; *Schmidt, Eisenhut & Schneider, 2020*). These processes have been widely studied in the plant model *Arabidopsis thaliana* (*Lequeux et al., 2010*; *De Smet et al., 2015*; *Niu et al., 2019*). Cu deficiency and excess cause imbalances in normal plant development and function (*Pätsikkä et al., 2012*; *Demirevska-Kepova et al., 2004*). The most common symptoms of Cu toxicity in plants are primary root length inhibition, vegetative tissue chlorosis, and the imbalanced capture of other relevant trace elements such as Fe, Mn, and Zn (*Lequeux et al., 2010*; *Andrés-Colás et al., 2013*).

In order to fully understand the homeostasis of Cu and other transition metals in plants, the role of plant-associated microorganisms must be considered (*González-Guerrero et al., 2016*; *Wasai & Minamisawa, 2018*). Soil microorganisms affect trace metal speciation, mobility, and availability in the soil matrix. Processes such as chemical transformation, chelation, and protonation increase the mobilization of trace elements, while precipitation and sorption decrease trace element availability (*Gadd, 2004*; *Mohamad et al., 2012*; *Rajkumar et al., 2012*; *Sessitsch et al., 2013*; *Muehe et al., 2015*). However, the role of metal-tolerant soil microorganisms in plant growth is still poorly understood, and additional studies using model microorganisms are clearly needed.

One suitable microorganism is *Cupriavidus metallidurans* strain CH34, a well known multi metal-resistant bacterial model that harbors at least 24 metal resistance gene clusters distributed acros four replicons: one chromosome (CHR1), one chromid (CHR2), and two megaplasmids (pMOL28 and pMOL30) (*Janssen et al., 2010*). This bacterium belongs to the *Burkholderiaceae* family, which is comprised of members that are capable of inhabiting diverse niches accross sludges, sediments, soils, rhizospheres, and plant roots (*Pérez-Pantoja et al., 2012*). Strain CH34 can withstand the millimolar range concentrations of heavy metal ions such as $Cu^+$, $Cu^{2+}$, $Ni^{2+}$, $Zn^{2+}$, and $Co^{2+}$, all of which are important for plant nutrition (*Mergeay et al., 2008*; *Kirsten et al., 2011*). Metal tolerance is mainly achieved by ion efflux, but also by metal complexation and reduction (*Mergeay et al., 2003*). The role of

this strain as a potential plant growth promoting rhizobacterium (PGPR) deserves further study.

In this study, we describe, for the first time, the effects of this multi metal-resistant strain on plant growth parameters, and the uptake and mobilization of metals (including several trace metal plant nutrients) in *A. thaliana* individuals that are exposed and not exposed to Cu. We additionally explored strain CH34's colonization of *A. thaliana*'s rhizosphere under these conditions and the transcriptional responses of a few specific genes of *C. metallidurans* CH34 in an *A. thaliana*-bacterium co-cultivation system.

## MATERIALS AND METHODS

### Bacterial strains, growth conditions, and *A. thaliana* inoculation

*C. metallidurans* CH34 was obtained from the Deutsche Sammlung von Mikroorganismen und Zellkulturen GmbH (Braunschweig, Germany). It was routinely grown in Dorn minimal saline medium (*Dorn et al., 1974*) containing 10 mM gluconate as a sole carbon and energy source in an orbital shaker (150 rpm) at 30 °C. We collected suspensions of cells that were fully grown to the mid-exponential phase ($OD_{600 nm} = 0.6$), adjusted them to approximately $10^8$ colony-forming units per milliliter ($CFU\ mL^{-1}$), and diluted them in agar medium just prior to solidification in order to obtain $10^4\ CFU\ mL^{-1}$ for *A. thaliana* inoculation of a gnotobiotic system. CFU were calculated based on colony counts from serial one mL dilutions (usually $10^{-2}$ to $10^{-8}$) of 100-μl aliquots plated on three R2A agar plate replicates. Ultimately, the CFU values corresponded to the average of colony counts x corresponding dilution $\times$ 10 (100 aliquot/1,000 uL dilution volume). To assess the effect of heat-inactivated bacteria, we heated an inoculum suspension at 95 °C for 20 min prior to the final dilution in agar medium. This temperature was high enough to kill all bacterial cells without destroying them (*Poupin et al., 2013*). Bacterial cell viability was routinely confirmed using R2A medium agar plate counting.

### In vitro *A. thaliana*-bacteria co-cultivation assays

*A. thaliana* Col-0 seeds were obtained from the Arabidopsis Biological Resource Center (Ohio State University, Columbus, OH, USA). The seeds were surface sterilized with 95% ethanol commercial laundry bleach (50%), rinsed three times with sterile water, and kept at 4 °C for 4 days to synchronize germination. Sterility was demonstrated in R2A or Luria Bertani media agar plates inoculated with aliquots of the solution used to wash seeds. Square Petri dishes were prepared with half-strength Murashige-Skoog medium ($MS^1/_2$) (*Murashige & Skoog, 1962*), 0.8% agar, and 1.5% sucrose (a carbon source for the plant, not for strain CH34); were either inoculated or not inoculated with strain CH34; and were supplemented with 0, 25, 50, 60, and 70 μM $CuSO_4x5H_2O$. The Petri dishes were then vertically placed in a growth chamber and incubated at 20−22 °C under a 16:8 h (long day) light:dark cycle. It should be noted that the copper concentration in the MS medium was 0.1 μM. We measured the *A. thaliana* growth parameters 21 days after sowing (DAS) to ensure that the parameters and bacterial colonization tests were accurate (*Poupin et al., 2013*; *Zúñiga et al., 2017*). The rosette areas were calculated using Adobe® Photoshop® Cs3 software (Adobe® Systems Incorporated, San Jose, CA, USA). We measured the

primary root growth using a ruler and counted the secondary roots. Fresh weight (FW) and dry weight (DW) were determined using a Shimadzu analytical balance (Shimadzu Corporation, Kyoto, Japan).

## Rhizosphere and *A. thaliana* colonization

To quantify bacterial colonization, we removed *A. thaliana* individuals at 21 DAS from inoculated agar that was either supplemented or not supplemented with 50 $\mu$M CuSO$_4$x5H$_2$O, and used vortex agitation to wash roots in a 10 mM MgSO$_4$x7H$_2$O solution in order to release their attached bacteria cells. The copper concentration was chosen based on its relation with positive or negative effects on *A. thaliana*. The extracted liquid was serially diluted with the same solution before plating on R2A agar plates, and the CFU mg$^{-1}$ FW values were determined after 24 h of incubation at 30 °C. To quantify rhizosphere colonization, we used vortex agitation to wash the roots in a 10 mM MgSO$_4$x7H$_2$O solution in order to release the samples of agar (MS$^1$/$_2$medium) that were in close contact with the roots (rhizosphere agar at a distance of less than one cm) and attached bacteria cells. We used *A. thaliana* plants that were grown without strain CH34 inoculation to check the sterility of the conditions. Sterility was proved after inoculation of aliquots from the solution used to wash samples of the agar material attached to roots, in rich media agar plates as indicated above. Agar plates without *A. thaliana*, and either supplemented with or without 50 $\mu$M CuSO$_4$x5H$_2$O, were used to determine *A. thaliana*'s influence on bacterial growth. Agar material samples were also washed with a 10 mM MgSO$_4$x7H$_2$O solution.

## Metal quantification in *A. thaliana* tissues

For total metal quantification, we separated the rosettes and roots of *A. thaliana* individuals 21 DAS, cultivated in the presence of 50 $\mu$M CuSO$_4$x5H$_2$O, and either inoculated or not inoculated with strain CH34, and rinsed them six times with ultra pure water. After drying for 24 h at 50 °C, the rosettes and roots material was separated into three 8 mg DW groups and digested using a HNO$_3$:H$_2$O$_2$ = 2:1 mixture in a high-performance microwave (Milestone, Ethos One, Brondby, Denmark). Metal analysis was performed using an ICP-MS Thermo Scientific X Series 2 spectrometer (ThermoFisher Scientific, Waltham, MA, USA). We calculated the translocation factor (TF) for each metal as the ratio between the rosette metal g$^{-1}$ $\mu$g and the root metal g$^{-1}$ $\mu$g, for each condition. Metal contents were determined by multiplying each sample concentration by the DW.

## Bacterial gene expression tests in *A. thaliana*-bacterium co-cultivation assays

To evaluate *C. metallidurans* CH34 gene expression in the presence of *A. thaliana*, we performed co-cultivation assays in hydroponic cultures. In order to allow adequate *A. thaliana* growth, 60 previously sterilized seeds were sown in the sterile conditions described above: MS (100%) medium supplemented with 3% sucrose, and 0 or 25 $\mu$M CuSO$_4$x5H$_2$O. This Cu concentration was chosen again because no detrimental effects were observed in the other *A. thaliana* individuals. *A. thaliana* growth was carried out under the same conditions as the in vitro cultures. At 21 DAS, the hydroponic cultures were inoculated with $1 \times 10^4$ cells/mL of strain CH34. We interrupted the *A. thaliana*-bacterium interaction

using quenching buffer (methanol 60%, 62.5 mM HEPES, and distilled water) after 30, 60, and 180 min of inoculation, and stored the biological material at $-80\,^{\circ}$C. These time periods were chosen because *A. thaliana* already influences bacterial gene expression (*Zúñiga et al., 2013*; *Zúñiga et al., 2017*). We used the same hydroponic systems that were inoculated with strain CH34 without *A. thaliana*, along with a growth medium containing gluconate supplemented with 0 or 25 µM CuSO$_4$x5H$_2$O, as the control to verify the effects on the induction and repression of bacterial genes in the absence of *A. thaliana* or Cu.

## Quantitative real time polymerase chain reaction (qRT-PCR) analysis

To extract RNA, we processed 4 ml of each co-cultivation assay using the geneJET RNA purification kit (ThermoFisher Scientific, Waltham, MA, USA). The RNA was quantified using an EON$^{TM}$ microplate reader (BioTek$^{®}$, Winoosky, VT, USA) and treated with the TURBO DNase kit (Ambion, Austin, TX, USA) to remove DNA contamination. cDNA synthesis was performed using the ImProm-II$^{TM}$ Reverse Transcription System (Promega Corporation, Madison, WI, USA) with 1 µg of RNA in 20 µL reactions. RT-PCR was performed using the Brilliant SYBR$^{®}$ Green QPCR Master Reagent kit (Agilent Technologies, Santa Clara, CA, USA) and the Eco$^{TM}$ Real-Time PCR detection system (Illumina$^{®}$, San Diego, CA, USA). The PCR mixture (10 µL) contained 2 µL of template cDNA (diluted 1:10) and 100 nM of each primer. Amplification was performed under the following conditions: 95 $^{\circ}$C for 10 min; followed by 40 cycles of 94 $^{\circ}$C for 30 s, 60 $^{\circ}$C for 30 s, and 72 $^{\circ}$C for 30 s; followed by a melting cycle from 55$^{\circ}$ to 95 $^{\circ}$C. Relative gene expression calculation was conducted as described in the software manufacturer's instructions (Illumina$^{®}$, San Diego, CA, USA). We calculated an accurate ratio between the expression of the gene of interest (GOI) and the housekeeping (HK) gene (16S rRNA) according to 2–($\Delta$CtGOI-HK). Firstly, we normalized using a ratio between the GOI and the 16S gene (as HK). Then, we used the time 0 treatment (before any treatment with MS or plant exudates) as the calibrating condition. The previous allows us to compare each treatment data with the others. It should be mentioned that the use of 16S rRNA is only for preliminary and explorative purposes and subsequent studies should include more appropriate housekeeping genes. Gene expression levels were normalized to the average expression level values in the control treatment. We also used reported primer pairs for *copK*, *copC*, *copC* 2, and *copF* gene sequences (Table 1), which were involved in the copper resistance of strain CH34 (*Monchy et al., 2006*). Primer pairs for *aleO*, *iucA*, *aleB*, *piuA*, *phaC1*, *catA*, *pcaG*, *bvgA,* and *phcA* genes (Table 1), involved in potential bacterial-plant interactions, were designed using Primer 3 version 0.4.0 (http://primer3.ut.ee). All PCR determinations were performed using at least three biological replicates and two technical replicates per treatment.

## Statistical analysis

The statistical analyses of *A. thaliana* growth parameters were performed using the two-way analysis of variance (ANOVA). Tukey's honest significant difference ($P < 0.05$) test was used to make comparisons across different treatments.

**Table 1  Sequences of primer pairs designed for this work.**

| Gene | Locus tag | Forward primer | Reverse primer |
|---|---|---|---|
| *aleO* | Rmet_5746 | GGAAGACGTCTATCGGACCA | CTGTGGCGTAACAGGATGTC |
| *iucA* | Rmet_1115 | GCATGTCGTGGATTGATCTG | GTGCAAGGGATATGGCTCAG |
| *aleB* | Rmet_1118 | ACGAAGAAGGACACGGTCAC | GGGAGATGGCTTGTGTTGTT |
| *piuA* | Rmet_4617 | TCTCGACGATTTCACGAATG | GCGGTCTCGTCATTGATCTT |
| *phaC1* | Rmet_1356 | ACCAATACGACGGCAGAGAG | ACTGAAACGGTTGGGACTTG |
| catA | Rmet_4881 | CGCTGGTAAAGCAGTTCCTC | TTGAACAGATCGGTCGTCAG |
| *pcaG* | Rmet_4014 | GTCGTTCCGCTTCGTTACC | GCATGAACACCGACACCAT |
| *bvgA* | Rmet_5714 | CCATTGGAGACTCGCTCTTC | GTCACGACCTTGACCTGCTT |
| *phcA* | Rmet_2977 | GATGCTGGGCCTGAATATGT | AGAAACTCGCCAACCACCT |

## RESULTS

### *C. metallidurans* CH34 effects on *A. thaliana* growth with varied Cu concentrations

We carried out in vitro *A. thaliana*-bacterium co-cultivation assays using a $MS^{1}/_{2}$ medium supplemented with 0 to 70 $\mu$M $Cu^{2+}$ over a 21-day period in order to test the effects of strain CH34 on growth. These $Cu^{2+}$ concentrations, particularly higher ones, caused mild to high stress in plants, reduced biomass, and increased chlorosis (*Lequeux et al., 2010*; *Andrés-Colás et al., 2013*). The presence of strain CH34 significantly altered *A. thaliana*'s rosette area, primary and secondary root growth, and FW and DW at all copper concentrations when compared with the non-inoculated group (Fig. 1A and 1B). When Cu concentrations were low to intermediate (0–50 $\mu$M), strain CH34 had a positive effect on *A. thaliana*, increasing its primary root length by 19.4%, 53.7%, and 19.0% in treatments with 0, 25, and 50 $\mu$M $Cu^{2+}$, respectively. The number of secondary roots also increased by 26.4% at 25 $\mu$M $Cu^{2+}$, and the rosette area increased by 10.3% and 21.0% at 0 and 25 $\mu$M $Cu^{2+}$, respectively. These changes in plant growth parameters appear to be modest, at first instance, but truly reflect effects of the presence of a single bacterium population on the plant.

However, at intermediate to high $Cu^{2+}$ concentrations (50–70 $\mu$M), strain CH34 had detrimental effects on primary root length, decreasing it by 20.0 and 86% in treatments with 60 and 70 $\mu$M, respectively, when compared with the non-inoculated control group. Rosette areas were reduced by 53.0, 60.0, and 75.3% at 50, 60, and 70 $\mu$M of $Cu^{2+}$, respectively. FW and DW decreased in all inoculated treatments, except at 25 $\mu$M $Cu^{2+}$ concentration, when compared to the non-inoculated condition (Fig. 1B). The decrease in FW and DW correlated with decreases in other *A. thaliana* growth parameters (Fig. 1B), except for root growth at 0 days, which might be explained by the initial reallocation of *A. thaliana* root resources upon bacterial inoculation. Both beneficial and detrimental effects were clearly caused by live CH34 cells, as the tests performed with heat-killed cells showed identical results as those of the non-inoculated control group (Supplementary Material, Fig. S1).
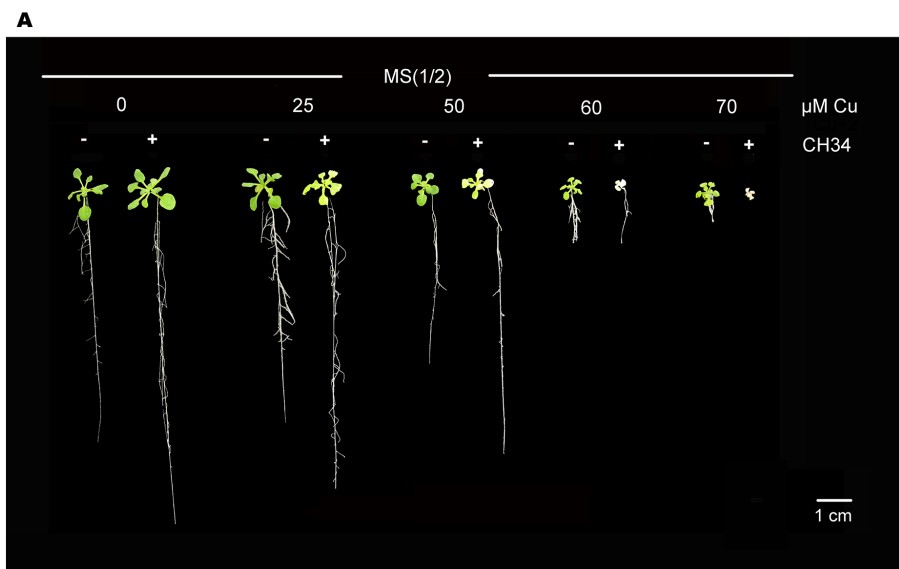

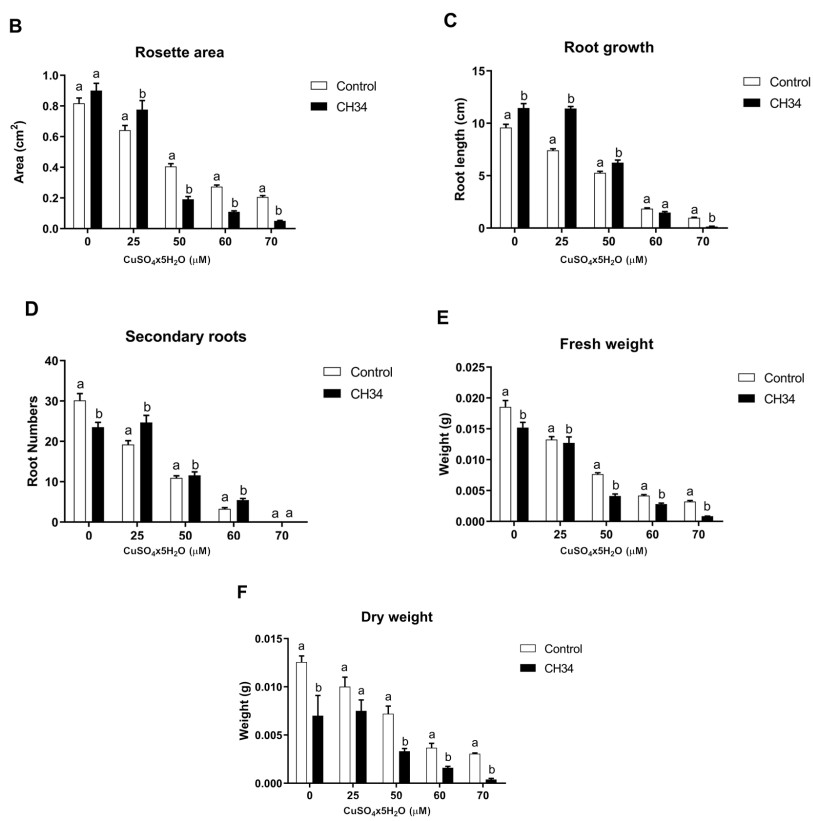

**Figure 1   Effect of *Cupriavidus metallidurans* CH34 on growth parameters of *Arabidopsis thaliana*.**
Rosette area, primary root growth, secondary roots and fresh and dry weights of *A. thaliana* Col-0 grown in in vitro gnotobiotic cultures using half strength Murashige-Skoog (MS) agar medium, inoculated with $1 \times 10^4$ colony forming units per milliliter of C. *metallidurans* CH34. Forty plants per condition were inoculated or not at day 1 and the half strength MS medium (continued on next page...)

**Figure 1 (…continued)**
was supplemented with 0 to 70 μM CuSO$_4$x5H$_2$O. (A) Representative control and strain CH34 inoculated plants grown in the presence of different Cu concentrations. (B) Plant growth parameters registered after 21 days of sowing. Bars show mean percentage values, and error bars indicate standard deviations from experiments with 30 plants analyzed for each bacterial condition. Different letters indicate statistically significant differences among treatments at each copper concentration for each measured parameter (two way ANOVA Tukey's HSD tests; $p < 0.05$).

### In control and Cu-exposed rosettes and roots, *C. metallidurans* CH34 modifies the accumulation and translocation of metals present in the *A. thaliana* growth medium

Cu may affect *A. thaliana* growth parameters (Fig. 1) via metal mobilization mediated by *C. metallidurans* (*Nies, 2016*). Therefore, we quantified the rosette and root tissue distribution of the main seven metals that composed the MS medium. Although not accurate, for simplicity we refer to boron as a metal. The metals and their concentrations were optimized for plant growth in the MS medium (*Murashige & Skoog, 1962*). However, when present in higher levels, these seven metals are toxic plant growth inhibitors (*Andresen, Peiter & Küpfer, 2018*). In inoculated *A. thaliana*, the absence of Cu caused a significant increase in B (29 and 32%), Co (68 and 67%), Cu (11 and 18%), Fe (42 and 91%), and Mn (11 and 46%) levels in rosettes and roots, respectively. However, Mo levels decreased (61%) and Zn levels increased (38%) in roots (absolute and relative values in Tables 2 and 3, respectively). In inoculated *A. thaliana*, the presence of Cu$^{2+}$ significantly increased Co (40 and 73%), Cu (55 and 38%), Mn (33 and 10%), Mo (60 and 64%), and Zn (21 and 71%) levels in rosettes and roots, respectively. B (30%) and Fe (9%) increased in rosettes only, Fe (31%) decreased in roots, and B levels in roots were not affected (Tables 2 and 3). In non-inoculated *A. thaliana*, 50 μM Cu$^{2+}$ increased B (109 and 380%), Co (50 and 20%), Cu (315 and 493%), Fe (277 and 315%), Mn (33 and 59%), and Zn (54 and 30%), and decreased Mo (27% and 70%) levels in rosettes and roots, respectively (Tables 2 and 3). To evaluate the metal mobility of *A. thaliana* roots and rosettes, TF values were determined for each metal (Table 4). In non-inoculated *A. thaliana*, we found that Cu clearly altered metal mobility because in the absence of Cu$^{2+}$, the TFs for Co (1.5-fold) and Mo (2.3-fold) increased, and the TFs for Cu (0.7-fold), Mn (0.8-fold), and B (0.4-fold) decreased. Strain CH34 innoculation in the absence of Cu changed the TFs of all elements except for B and Co, but in the presence of Cu, the strain modified all TFs except for Mo (Table 4). In the absence of strain CH34, Cu$^{2+}$ drastically changed the metal accumulation in *A. thaliana* roots and rosettes, with much higher B, Cu, and Fe accumulation; mild Co, Mn, and Zn accumulation; and negative (less accumulation in the presence of 50 μM Cu$^{2+}$) Mo accumulation (Table 3).

### Copper-stressed *A. thaliana* changed *C. metallidurans* CH34 colonization and the transcriptional levels of some genes

To further explore Cu's effects on the *A. thaliana*-*C. metallidurans* interaction, we conducted rhizospheric and colonization tests in the presence of 50 μM Cu$^{2+}$. This concentration was chosen because it provoked intermediate effects on *A. thaliana* growth parameters. *C. metallidurans* effectively colonized ($1 \times 10^8$ CFU mL$^{-1}$) both *A. thaliana*
**Table 2  Metal levels in rosettes and roots of *Arabidopsis thaliana* 21 days after sowing, inoculated or not with *Cupriavidus metallidurans* CH34 in the presence or absence of copper.**

| | Rosettes | | | | Roots | | | |
|---|---|---|---|---|---|---|---|---|
| | $MS^1/_2$ | | $MS^1/_2 + 50\,\mu M\ Cu^{2+}$ | | $MS^1/_2$ | | $MS^1/_2 + 50\,\mu M\ Cu^{2+}$ | |
| | Control | +CH34 | Control | +CH34 | Control | +CH34 | Control | +CH34 |
| B | $21.0 \pm 1.6^{a**}$ | $27.5 \pm 2.2^{b}$ | $44.0 \pm 3.4^{c}$ | $57.3 \pm 5.3^{d}$ | $14.6 \pm 2.0^{a}$ | $19.3 \pm 0.9^{b}$ | $70.1 \pm 23.4^{c}$ | $59.2 \pm 23.9^{c}$ |
| Co | $0.2 \pm 0.0^{a}$ | $0.3 \pm 0.0^{b}$ | $0.3 \pm 0^{c}$ | $0.5 \pm 0.0^{d}$ | $0.5 \pm 0.1^{a}$ | $0.9 \pm 0.0^{b}$ | $0.6 \pm 0.1^{a}$ | $1.0 \pm 0.1^{b}$ |
| Cu | $4.5 \pm 0.0^{a}$ | $4.7 \pm 0.2^{b}$ | $186.9 \pm 6.1^{c}$ | $289.0 \pm 1.4^{d}$ | $5.2 \pm 0.1^{a}$ | $6.2 \pm 0.3^{b}$ | $308.5 \pm 13.8^{c}$ | $426.2 \pm 20^{d}$ |
| Fe | $118.0 \pm 5.4^{a}$ | $167.9 \pm 9.0^{b}$ | $445.0 \pm 34.4^{c}$ | $484.1 \pm 11.5^{d}$ | $2671.6 \pm 240.8^{a}$ | $5091.6 \pm 180.0^{b}$ | $11098.0 \pm 336.9^{c}$ | $7636.7 \pm 377.7^{d}$ |
| Mn | $196.5 \pm 9.7^{a}$ | $217.7 \pm 12.4^{b}$ | $262.6 \pm 15.4^{c}$ | $350.4 \pm 7.2^{d}$ | $111.3 \pm 9.0^{a}$ | $163.0 \pm 4.1^{b}$ | $177.1 \pm 9.7^{c}$ | $195.2 \pm 12.2^{d}$ |
| Mo | $19.5 \pm 1.4^{a}$ | $18.4 \pm 2.1^{a}$ | $14.3 \pm 0.4^{b}$ | $23.0 \pm 1.2^{c}$ | $66.1 \pm 7.3^{a}$ | $25.8 \pm 4.7^{b}$ | $21.3 \pm 4.1^{c}$ | $35.0 \pm 1.6^{b}$ |
| Zn | $191.4 \pm 17^{a}$ | $191.8 \pm 16.5^{a}$ | $295.7 \pm 15.1^{b}$ | $359.2 \pm 1.5^{c}$ | $940.4 \pm 56.2^{a}$ | $1299.2 \pm 55.9^{b}$ | $1222.8 \pm 46.7^{c}$ | $2091.8 \pm 107.8^{d}$ |

**Notes.**

[**] All values are expressed as $\mu$g metal $g^{-1}$ of dry weight.

Standard Murashige-Skoog plant growth medium. $MS^1/_2 + 50\,\mu M\ Cu^{2+}$ indicates same medium supplemented with 50 $\mu$M $CuSO_4x5H_2O$. Each condition had three technical replicates consisting of at least 8 mg of rosette or root material. Different letters represent significant differences between rosette or root levels of each metal (two-way ANOVA, $p < 0.05$; Tukey test, $p < 0.05$).

**Table 3** Percentage of metal accumulation in rosettes and roots of *Arabidopsis thaliana* 21 days after sowing, inoculated or not with *Cupriavidus metallidurans* CH34, in the presence or absence of copper.

| Element | Rosettes (CH34/control) | | Roots (CH34/control) | | Rosettes (Cu/control) | Roots (Cu/control) |
|---------|------------------|------------------------|------------------|------------------------|-----------------------|----------------------|
| | $MS^1/_2$ | $MS^1/_2 + 50\,\mu M\ Cu^{2+}$ | $MS^1/_2$ | $MS^1/_2 + 50\,\mu M\ Cu^{2+}$ | | |
| B | +29%[**] | +30% | +32% | – | +109% | +380% |
| Co | +68% | +40% | +67% | +73% | +50% | +20% |
| Cu | +11% | +55% | +18% | +38% | +315 | +493% |
| Fe | +42% | +9% | +91% | −31% | +277% | +315% |
| Mn | +11% | +33% | +46% | +10% | +33% | +59% |
| Mo | – | +60% | −61% | +64% | −27% | −69% |
| Zn | – | +21% | +38% | +71% | +54% | +30% |

Notes.

[**]All values were calculated as percentages of the metal level (µg of metal g$^{-1}$ of dry weight) in inoculated versus non-inoculated plants (CH34/control), and plants in the presence or absence of copper (Cu/control).
Not significant changes.

**Table 4** Translocation factors (TF) for each metal in *Arabidopsis thaliana* inoculated or not with *Cupriavidus metallidurans* CH34, in the presence or absence of copper.

| | $MS^1/_2$ | | $MS^1/_2 + 50\,\mu M\ Cu^{2+}$ | |
|---|------------------|------------------|------------------|------------------|
| | $TF_{Control}$ | $TF_{CH34}$ | $TF_{Control}$ | $TF_{CH34}$ |
| B | 1.6[a] | 1.3[a] | 0.7[b] | 1.1[b] |
| Co | 0.4[a] | 0.4[a] | 0.6[b] | 0.5[a] |
| Mo | 0.3[a] | 0.8[b] | 0.7[b] | 0.7[b] |
| Cu | 0.9[a] | 0.8[b] | 0.6[c] | 0.7[d] |
| Mn | 1.9[a] | 1.3[b] | 1.5[c] | 1.8[d] |
| Fe | 0.04[a] | 0.03[b] | 0.04[a] | 0.06[c] |
| Zn | 0.2[a] | 0.1[b] | 0.2[c] | 0.1[d] |

Notes.

TF was calculated as = (Metal concentration$_{Rosette}$/Metal concentration$_{Root}$). $MS^1/_2$ corresponds to half-diluted standard Murashige-Skoog plant growth medium. $MS^1/_2 + 50\,\mu M\ Cu^{2+}$ indicates same medium supplemented with 50 µM $CuSO_4x5H_2O$. Each condition had three technical replicates consisting of at least 8 mg of rosette or root material. Different letters represent significant differences between treatments (two-way ANOVA, $p < 0.05$; Tukey test, $p < 0.05$).

and its rhizospheric surroundings. Adding copper produced significant changes on the bacterial colonization by both increasing bacteria directly adhered to *A. thaliana* roots by 3.1 times and decreasing colonization in its vicinity by 7.1 times (Table 5). It should be noted that we observed positive effects on strain CH34 colonization as a 4-log increase in CFU with respect to the non-rhizospheric conditions was recorded (Table 5).

To further understand the effects of strain CH34 on *A. thaliana* protection, colonization, and metal translocation, we carried out a transcriptional analysis of a small subset of genes. These experiments were performed in *A. thaliana* hydroponic cultures in order to fulfill the quality requirements for bacterial RNA extraction. Since Cu has a greater presence in hydroponic medium than in agar plates (*Burkhead et al., 2009*), we used a 25 µM $Cu^{2+}$ concentration to emulate the 50 µM $Cu^{2+}$ used in agar plate *A. thaliana* tests in order to provoke similar effects. At 21 days, *C. metallidurans* was inoculated and RNA was extracted after 30, 60, and 180 min from the hydroponic cultures. These bacteria—*A.*

**Table 5** Colony forming units of *Cupriavidus metallidurans* CH34 by milligram of fresh weight (CFU mg$^{-1}$ FW) of plant, agar surrounding plant (rhizosphere), and non-planted MS$^1/_2$ agar medium (non-rhizosphere).

| Treatment | Plant (CFU mg$^{-1}$FW) | Rhizosphere (CFU mg$^{-1}$FW) | Non-rhizosphere (CFU mg$^{-1}$FW) |
|---|---|---|---|
| MS($^1/_2$) | $5.8 \times 10^8 \pm 4.7 \times 10^{8}$[a] | $1.0 \times 10^7 \pm 8.5 \times 10^{6}$[a] | $1.7 \times 10^4 \pm 1.1 \times 10^{4}$[a] |
| MS($^1/_2$) + 50 μM Cu$^{2+}$ | $1.8 \times 10^9 \pm 7.5 \times 10^{7}$[a] | $1.4 \times 10^6 \pm 1.2 \times 10^{6}$[b] | $3.6 \times 10^4 \pm 1.3 \times 10^{4}$[b] |

**Notes.**

*C. metallidurans* CH34 CFU mg$^{-1}$ FW were counted in 21 days after sowing plants, rhizosphere agar and non-rhizosphere MS$^1/_2$ agar, in 0 and 50 μM CuSO$_4$x5H$_2$O treatments. Plants were initially inoculated with $1.0 \times 10^4$ CFU ml$^{-1}$. The average values of 15–20 replicates and their respective standard deviations are shown. FW corresponds to that of plant or agar material. Different letters in same column indicate statistically significant differences according to a t-test, $p < 0.05$.

*thaliana* interaction times were selected based on a previous study that used a similar *Burkholderiaceae* bacterium (*Zúñiga et al., 2017*). Three groups of genes were chosen to perform a RT-qPCR analysis: a set of *cop* genes involved in Cu detoxification, a set of siderophore-related genes, and a set of colonization-related genes (*Janssen et al., 2010*). Figure 2 shows the relative expression levels of the genes associated with copper resistance: *copK*, *copF*, *copC*, and *copC2*. The first three genes are located in the megaplasmid pMOL30 in a cluster of 21 *cop* genes and they encode proteins involved in periplasmic (*copK* and *copC*) and cytoplasmic (*copF*) detoxification. *copC2*, located on a smaller cluster of *cop* genes (*copD$_2$C$_2$B$_2$A$_2$R$_2$S$_2$*), is strongly induced mainly at concentrations lower than 100 μM Cu$^{2+}$ (*Janssen et al., 2010*). *copK* was clearly expressed 60 min after *A. thaliana* inoculation (Fig. 2), with induction still observable after 180 min in the presence, but not in the absence, of Cu$^{2+}$. *copC* only showed a transient significant induction in the presence of Cu$^{2+}$. *copF* gene expression levels remained essentially unchanged and *copC2* expression levels noticeably decreased in the presence of *A. thaliana* (Fig. 2).

A second set of targeted genes were involved in the regulation, production, and sensing of *C. metallidurans* CH34 staphyloferrin B siderophore. A previous study found that *C. metallidurans* CH34 affects metal availability by producing a unique phenolate-type siderophore that has been described to bind Cd$^{2+}$ (*Gilis et al., 1998*). The Fur protein (*aleO* gene) is a transcriptional regulator involved in Fe metabolism that represses staphyloferrin B synthesis under optimal Fe conditions, although it has been suggested that the transcriptional activation of other genes is also involved in Fe regulation (*Ma, Jacobsen & Giedroc, 2009*). After 30 and 180 min of *C. metallidurans* CH34 inoculation, Fur regulator expression was significantly repressed in both *A. thaliana*-only and *A. thaliana* plus 25 μM Cu$^{2+}$-conditions, suggesting that Fe requirement should not be as severe as in 25 μM Cu$^{2+}$-only conditions (Fig. 3). The *iucA* gene, one of the three genes responsible for staphyloferrin B synthesis, was only significantly induced in 25 μM Cu$^{2+}$-only conditions (Fig. 3), while the staphyloferrin B receptor *aleB* gene was induced after 60 and 180 min of inoculation in the presence of *A. thaliana*. The *piuA* gene encoding a siderophore/iron-complexation receptor was not involved in staphyloferrin B uptake, and was only induced in the 25 μM Cu$^{2+}$-only condition (Fig. 3). Finally, we analyzed the expression changes in *phaC1* (polyhydroxybutyrate synthesis), *catA* (catechol), *pcaG* (protocatechuate catabolism), *bvgA* (transcriptional activator of virulence and colonization factors), and *phcA* (virulence regulator) genes. *phaC1* was the only gene that showed clear induction by *A. thaliana* (peaked at 30 min, Supplementary Material Fig. S2), as the other

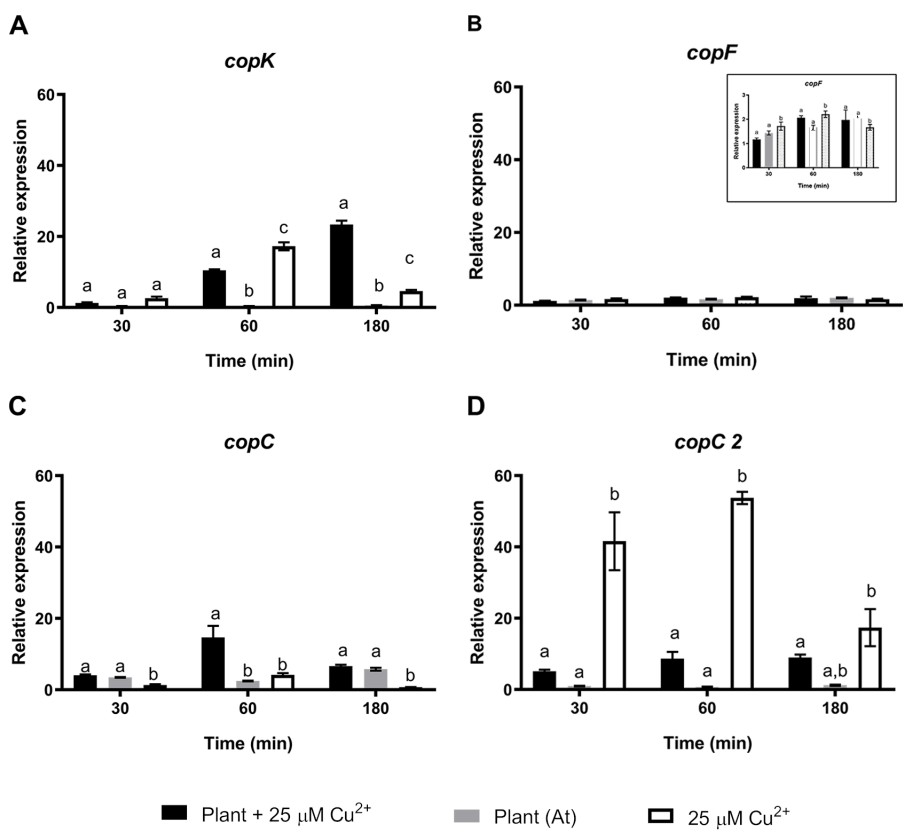

**Figure 2** Relative expression levels of Cu resistance genes from *Cupriavidus metallidurans* CH34, in the presence or in the absence of *Arabidopsis thaliana (At)* or copper. Quantitative Real Time Polymerase Chain Reactions determinations of relative expression levels of (A) *copK*, (B) *copF*, (C) *copC*, and (D) *copC 2* genes. Expression levels were normalized respect to the housekeeping gene 16S rRNA. Data correspond to means ± standard deviations of at least three biological replicates. Different letters indicate significant differences between same time conditions (two way ANOVA Tukey's HSD tests; $p < 0.05$).

four tested genes were either not induced, or only poorly induced, by *A. thaliana*. Since the same medium was used across all tests, *phaC1* gene's changes could not be explained by phosphate changes.

## DISCUSSION

Our results indicate that the *C. metallidurans-A. thaliana* interaction is characterized by a dual effect on plant growth and metal availability. A dependence on $Cu^{2+}$ levels, and significant effects on of *A. thaliana*'s bacterium colonization and bacterial gene expression. We have previously demonstrated that closely related species such as *C. pinatubonensis* and *P. phytofirmans* have also PGPR abilities (*Ledger et al., 2012*; *Poupin et al., 2013*; *Zúñiga et al., 2017*) but not showing this dual effect.

*C. metallidurans*' ability to act as a PGPR (see the increase in root length at low $Cu^{2+}$ levels, Fig. 1), at least in *A. thaliana*, is not surprising as several members of the *Cupriavidus* genus and the *Burkholderiaceae* family are well-known PGPRs or, at least,

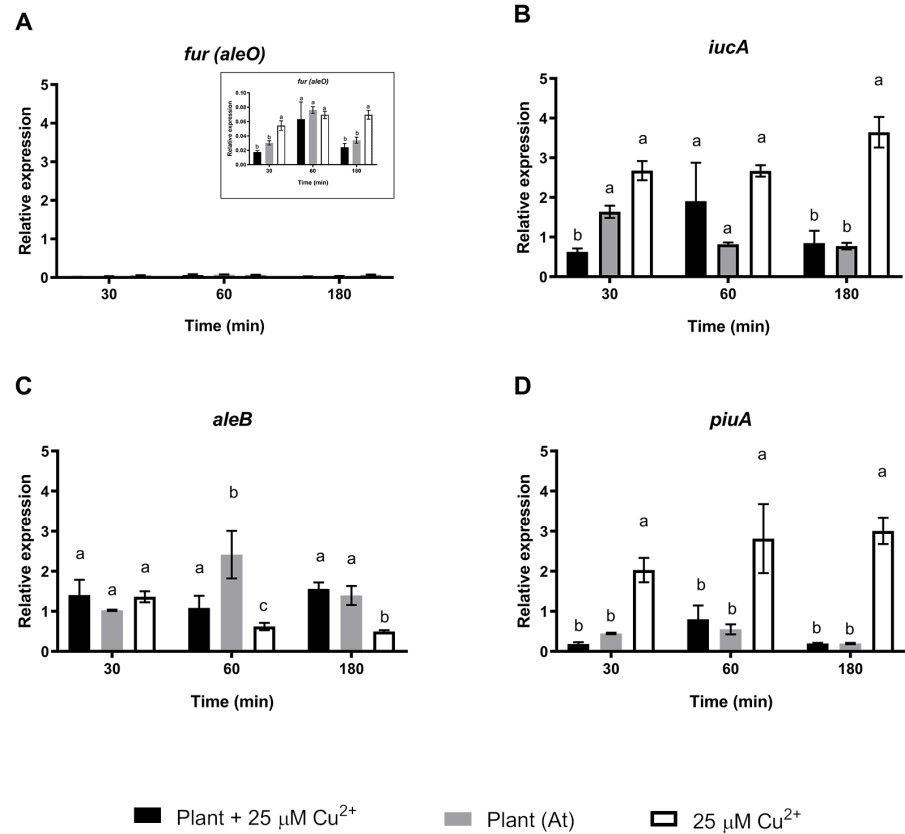

**Figure 3** **Relative expression levels of staphyloferrin B siderophore production genes from *Cupriavidus metallidurans* CH34, in the presence or in the absence of *Arabidopsis thaliana (At)* or copper.** Quantitative Real Time Polymerase Chain Reactions determinations of relative expression levels of (A) *fur (aleO)*, (B) *iucA*, (C) *aleB*, and (D) *piuA* genes. Expression levels were normalized respect to the housekeeping gene 16S rRNA. Data correspond to means ± standard deviations of at least three biological replicates. Different letters indicate significant differences between same time conditions (two way ANOVA Tukey's HSD tests; $p < 0.05$).

interact with plants (*Gyaneshwar et al., 2011*; *Bhattacharyya & Jha, 2012*; *Poupin et al., 2013*; *Arroyo-Herrera et al., 2020*). Because PGPR traits (*Zeffa et al., 2020*), including 1-aminocyclopropane-1-carboxylate deaminase, auxin phytohormone production, phosphate solubilization, and nitrogen fixation (*Bhattacharyya & Jha, 2012*; *Aeron et al., 2020*), have not yet been reported in *C. metallidurans*, new PGPR traits would be interesting to explore in this bacterium. New PGPR traits could emerge from further in-depth study of the responses that gobernate this interkingdom relationships. Furthermore, the colonization behavior reported here is indicative of the inherent dynamics of the rhizospheric environment (*Bais et al., 2006*; *Chaparro, Badri & Vivanco, 2014*) and highlight the importance to describe and study these traits in environmental intertwined plant-microbre systems.

Few studies have addressed the effects that *C. metallidurans* CH34 exerts in plants and their rhizosphere environments. Because it had no apparent plant growth-promoting or

protection effects, strain CH34 has been employed as a neutral bacterium (*Taghavi et al., 2009*). In a bioaugmentation-assisted phytoextraction procedure, strain CH34 was shown to increase Cr and Pb levels in maize shoots by a factor of 4.4 and 3.2, respectively, but it reduced Cr and Pb levels in roots by a factor of 2.9 and 4.8, respectively, illustrating how metal uptake and mobilization depend on metal species and plant tissue (*Braud et al., 2009*). Our results suggest a possible linkage between the plant metal uptake and the plant development when *C. metallidurans* CH34 is present. Microbially mediated plant growth promotion effects through metal level increases in Arabidopsis, have been scarcely explored (*Lu et al., 2020*).

The effects of negative Cu excess on plant growth have been previously studied (*Doncheva, 1998*; *Burkhead et al., 2009*; *Kumar et al., 2021*), and found that Cu mostly accumulates in root tissues (*Alaoui-Sossé et al., 2004*; *Navari-Izzo et al., 2006*; *Burkhead et al., 2009*). Cu provokes the reorganization of root architecture, leading to primary root growth inhibition and secondary root stimulation (*Lequeux et al., 2010*), although the latter effect has not been observed in *A. thaliana*, at least under the growth conditions used in this study. The detrimental effects of high Cu levels reported here may also be related to the uncontrolled mobilization of other micronutrients (*Andresen, Peiter & Küpfer, 2018*; *Kumar et al., 2021*).

It can be hypothesized that the dual effect of *C. metallidurans* on *A. thaliana* growth, either beneficial or detrimental depending on the Cu concentration, may be explained, although not exclusively, by the metal mobilization stimulated by this bacterium (Tables 3 and 4). This strain possesses transport mechanisms for multiple metallic ions, siderophores, and other metal-chelating molecules (*Kirsten et al., 2011*; *Nies, 2016*). At low Cu levels, strain CH34 may promote *A. thaliana* growth, increase micronutrient levels, and enhance uptake by roots and further mobilization to upper tissues. At high $Cu^{2+}$ concentrations, such enhanced uptake and mobilization processes may cause micronutrient tissue accumulation at sub-toxic or toxic levels, impairing *A. thaliana* growth. It has been previously reported that Cu can alter the accumulation and translocation of transition metals (*Karimi et al., 2012*; *Andrés-Colás et al., 2013*), causing a reduction of Mn and Fe and an augmentation of Zn total content in roots. Under the conditions tested here, Cu's effects were especially significant for B and Fe accumulation, as well as Cu (*Puig et al., 2007*; *Puig & Peñarrubia, 2009*). Mo mobilization is clearly blocked, in a less clear process (*Andresen, Peiter & Küpfer, 2018*).

The discrepancies in our results with those of other studies can probably be explained by our testing of micronutrient concentrations that were lower than those used in other studies (*Doncheva, 1998*; *Alaoui-Sossé et al., 2004*; *Andrés-Colás et al., 2013*; *Saleem et al., 2020*). Additionally, our experimental approach involved the addition of six metals along with Cu, which has not been previously addressed in literature and has implications for the use of *C. metallidurans* in phytoremediation procedures for metal-containing polluted sites (*Gadd, 2004*; *Glick, 2010*; *Rajkumar et al., 2012*), especially those contaminated with metal mixtures. There have been several recent studies on the use of microorganisms to remediate sites polluted with metal combinations (e.g., *Cameron, Mata & Riquelme, 2018*; *Choinska-Pulit, Sobolczyk-Benarek & Laba, 2018*). Interestingly, co-inoculation approaches have

proved successful when protecting alfalfa from Cu/Zn mixtures (*Jian et al., 2019*), cleaning slurries containing heavy metal mixtures plus polycyclic aromatics (*Subashchandrabose et al., 2019*), or phytoremediating mine tailings from a Cu, Fe, Zn, and sulfur extraction operation (*Benidire et al., 2021*). Therefore, strain CH34 may be a good candidate to be used as a microbuial consortium member in co-inoculation schemes to take advantage of the beneficial properties and to prevent potential deleterious effects (Cillero et al. submitted, Supplementary Material).

The colonization of *A. thaliana* by strain CH34 cells indicated that in the presence of Cu, microenvironment conditions next to the plant increased the attraction and/or proliferation of this bacterium, which may be explained by changes produced in the composition of *A. thaliana* root exudates, mainly composed of sugars, amino acids, and aromatic compounds (*Sasse, Martinoia & Northern, 2018*; *Vives-Peris et al., 2020*). Strain CH34, as well as other closer *Cupriavidus* strains, is not able to grow on common sugars but it is able to proliferate on *A. thaliana* root exudates (Cillero et al. submitted, Supplementary Material). *Cupriavidus* strain genomes usually encode a significant group of genes encoding degradation of aromatic compounds that enable these strains to use a wide range of such organic compounds for growth (*Lykidis et al., 2010*; *Pérez-Pantoja et al., 2015*), providing ecologically relevant advantages, such as those reported in the closely-related strain *C. pinatubonensis* JMP134 (*Ledger et al., 2012*; *Pérez-Pantoja et al., 2015*). Additionally, Cu-induced stress in plants may also increase root exudation (*Huang et al., 2016*). Variations in the rhizosphere microenvironment have a mayor role modulating the rhizospheric microbiome, which is capable of recruit or discard some microbial species to dynamicly adjust benefic or detrimental interactions on the system (*Zhalnina et al., 2018*). *C. metallidurans* colonization features reported here may reflect system responses to these microenvironmental changes.

We wanted to start exploration of effects in strain CH34 gene expression of the presence /absence of the plant and Cu. The presence of *A. thaliana* and Cu differentially affected strain CH34 expression of some genes. For instance, *cop* genes involved in metal detoxification (*copK* and *copC*) were influenced by *A. thaliana* and Cu to a greater extent than the other *cop* genes, suggesting that periplasmic detoxification (*Monchy et al., 2006*) plays a significant role in the *A. thaliana*-bacterium-metal interaction. Genes involved in Fe turnover were also affected by interaction in the *A. thaliana*—*C. metallidurans* –copper triad, as has been previously reported (*Puig et al., 2007*). Plant-induced *phaC1* gene expression was also in accordance with *C. metallidurans*' interactions with *A. thaliana*, as polyhydroxybutyrate production is an advantage reported during early plant colonization (*Kadouri, Jurkevitch & Okon, 2005*; *Balsanelli et al., 2016*). Additional studies are clearly required as some no significant changes reported here for earlier times might not be found at later stages (7, or 14 days).

## CONCLUSION

This work contributes to our understanding of plant-microbe interactions. At least for the metal multi-resistant *C. metallidurans*—*A. thaliana* system, we established that the dual

(beneficial or detrimental) effect is dependent on copper levels, the metal root-to-shoots translocation is affected, and the validation of *A. thaliana* bacterial colonization and expression of some bacterial genes. Therefore, the role of metal-tolerant bacteria thriving in the rhizosphere of the plant-bacteria-metal triad deserves special attention, and plant growth promotion, protection, and/or phytoremediation strategies should be explored. Metal (Cu) levels play a key role and are probably the main factor controlling plant-microbe interactions. However, it should be kept in mind that the results reported here were derived under laboratory conditions. Further research is required to validate plant-metal tolerant bacterium-metal contents under field conditions.

## ACKNOWLEDGEMENTS

We thank Santiago Andrade for its assistance and technical advice on the use of ICP-MS equipment at Pontificia Universidad Católica de Chile.

### Funding

This work was supported by grants from FONDECYT 1151130 and 1190634, and ANID PIA/BASAL FB0002. The funders had no role in study design, data collection and analysis, decision to publish, or preparation of the manuscript.

### Competing Interests

The authors declare there are no competing interests.

### Author Contributions

- Claudia Clavero-León and Javier Cillero performed the experiments, analyzed the data, prepared figures and/or tables, and approved the final draft.
- Daniela Ruiz conceived and designed the experiments, performed the experiments, analyzed the data, prepared figures and/or tables, and approved the final draft.
- Julieta Orlando and Bernardo González conceived and designed the experiments, analyzed the data, authored or reviewed drafts of the paper, and approved the final draft.

### Data Availability

  Raw data are available as Supplemental Files.

### Supplemental Information

Supplemental information for this article can be found online at http://dx.doi.org/10.7717/peerj.11373#supplemental-information.

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
