# Peer review of "The multi metal-resistant bacterium Cupriavidus metallidurans CH34 affects growth and metal mobilization in Arabidopsis thaliana plants exposed to copper"

_PeerJ, doi:10.7717/peerj.11373_

## Round 0.1 · original submission · Major Revisions

The reviewer comments are quite clear. Many comments are related to grammar, so I would encourage you to focus on those after you have addressed the technical comments.

·

Basic reporting

• The objective of the manuscript is rather clear.
• English require verification (check by a native speaker)
• Intro & background should be further developed, at the moment too general.
• Literature well referenced & relevant, however more literature review for the discussion section is necessary.
• Structure conforms to PeerJ standards: Results and discussion should be separately following the journal’s requirements.
• Figures are definitely too small and need some improvement. Table are missing important information (see my specific comments).
• Raw data supplied, however, metadata not provided (description of columns’ names is needed)

Experimental design

• Original primary research within Scope of the journal.
• Research question well defined, relevant & meaningful.
• It is not clearly stated how the research fills an identified knowledge gap.
• Investigation performed to a rather low but sufficient technical standard.
• Some methods described with insufficient detail & information to replicate (see my specific comments).

Validity of the findings

• Moderate impact and novelty.
• Inconclusive results.
• All underlying data have been provided, however not always presented in a clear way.
• Data are robust, statistically sound, & controlled.
• Conclusions are not well stated, rather vague and general. Not enough linked to original results

Additional comments

• Authors should highlight the importance, novelty, and significance of this work.
• All the experiments were conducted in growth media and hydroponic cultures. Did the authors consider using more environmentally relevant matrix such as soil? That could significantly change the uptake and translocation of metals and certainly, it would give a better idea about the potential use of CH34 strain in bioremediations. It should be mentioned in the text…
• The plant root exudes (PRE) analysis is barely mentioned in the discussion and results are not presented in any form (figure/table). Whether authors further elaborate on this result or completely remove it. At the moment I don’t feel it contributes much to the discussion and interpretation of results.
• Generally, many sentences are written in a clumsy way. I strongly recommend giving the text for a revision to a native speaker.

Reviewer 2 ·

Basic reporting

The authors should thoroughly check the style and grammar of the complete manuscript (including the abstract.
Some examples: L21: “but it excess”?; L31: Capital of sentence; L54-60: very long sentence; L67: chromid instead of chromide; L75; L222: “live strain CH34 cells”, remove “strain”; L286: “The effects on metal levels reported here have”; L236: “because it provoked”; L305: “located on”.; L320: Cd2+

Experimental design

- Why are the results with heat-killed cells not shown?
- Experiments with colonization (Table 2): agar plates without plants were used as control but the data are expressed as CFU/mg FW. Furthermore, this shows a 4-log decrease (initial inoculum is 10^8 CFU/ml), which is not discussed.
- The data on changes in PRE composition are not shown in detail.
- It would be interesting to also add impact of Cu on metal accumulation (% increase or decrease) without CH34 in Table 4.
- qPCR: It is unclear if gluconate was included in both the plant and control conditions, which could have a great impact on expression levels. It would be interesting to use the delta delta ct method and compare expression in plants with or without Cu. In addition, was primer efficiency tested?
- L297-298: is there information available to justify the correlation between 50 µM in agar and 25 µM in hydroponics?
- L305 and further: for the copC on the chromid, please refer to copC2
- L309 and futher: copK was clearly expressed 60 min after inoculation, but only in the presence of Cu, right? (Figure 2); copF was not changed: but significantly different at 30 and 180 min (plants vs plants with Cu)? Please revise this section.
- L332: impact on phaC1 expression: Is there any difference in phosphate content of the medium in the experiments with and without plants? If so, this could have an impact on phaC1 expression and should be discussed.

Validity of the findings

- The differences between beneficial and detrimental effects based on Cu concentration should be reported and discussed much more nuanced in both the abstract and body of the text (results and discussion) as the impact on the different parameters. For instance, L202-205, however there is no effect on FW and DW at 25 µM. L208: 10.3% increase of rosette area at 0 µM, but not significant?; The cutoff at 50 µM (positive/negative effects) is not that clear cut.
- What is the evidence for the presumably new PGPR traits in CH34?
- The expression data concerning Fe and siderophores metabolism should also be discussed in light of the metal accumulation experiments, indicating increased Fe uptake (167 µg/DW without and 484 µg/DW with Cu) that could impact Fe availability for CH34.

Additional comments

The authors provide an interesting study on the impact of strain CH34 in the presence of absence of Cu on Arabidopsis. Although they show interesting findings, the should present them with more details and nuances.

Reviewer 3 ·

Basic reporting

The English language could be improved.
Some examples where the language could be improved or shown below:

in general: try to be consistent in the spelling of Cu: sometimes Cu, sometimes Cu(II) and sometimes Cu+2

Line 21: it excess -> in excess, it
Line 22: interacts -> interact
Line 25: on the growth of
Line 26: gene
Line 32: reveal the complexity
Line 33: on the use of
Line 51: are the inhibition
Line 52: capture of -> capturing
Line 54: metal -> metals
Line 54-59: very long sentence. Can be split in two or three sentences to make it more readable
Line 63: in the soil
Line 67: chromide -> chromid
Line 68: comprises
Line 83: In this work,
Line 121: add space between roots & 10 mM
Line 126: the sterile conditions? Sterility of the conditions?
Line 131: To obtain PRE,
Line 135: as for the
Line 137: one -> medium
Line 161: mL-1 cells? Cells/ml?
Line 196: on the Cu concentration
Line 198: to test the effects
Line 208: remove a
Line 236: because this concentration
Line 256-line 261: very long sentence. Can you rephrase to make it more readable?
Line 286: the effects on metals levels? -> the effect of metal levels?
Line 293: the effects
Line 304: shows the relative
Line 307: remove gene
Line 309: remove gene
Line 310: remove gene
Line 320: the Fur protein

The introduction gives a comprehensive overview of but could be improved by a more detailed description of the copper resistance mechanisms in strain CH34 as this is important in one of the results sections.

In general, the figures and tables are readable, but
Figure 2 legend: do you mean plants + CH34? All conditions are the result of a comparison Cu-induced or without Cu? To make a better comparison, it can help to put the Y-axes at the same scale so it is immediately clear that there is a big difference is relative expression levels of the different genes (e.g. almost no expression of copF)
Figure 3 is not readable. Can it be adapted to a larger format?

Experimental design

The research question from this paper is clear: the authors want to investigate whether CH34 can be beneficial for copper tolerance of Arabidopsis thaliana.

The experiment how rhizosphere and plant colonization was tested is not really clear to me. How was the agar treated? also rinsed with MgSO4?

I have a specific question regarding the gene expression: why did you use a different medium for the samples without plants? How will you discriminate for the differences due to the different medium?

Validity of the findings

Some general remarks:
As you mention in the discussion: CH34 is not able to metabolize sucrose. This you also see in the cell counts of the non-rhizosphere conditions, which remain 104 CFU/ml. Here, you also see that copper itself does not have negative impact on the growth or survival of CH34. It shows that CH34 needs the rhizosphere or plant environment to grow. As plants excrete more organic compounds due to the copper treatment, CH34 has maybe more carbon available to grow and that can explain the difference? Did you check if CH34 can metabolize the exudates? This could be elaborated more in the text. Maybe it is rather a matter of available energy sources to grow instead of a change in the microenvironment?
You did a sterility check but results are not shown? It is good (even in supplementary data) to prove that your samples are still sterile.
Regarding the effect on CH34 on the growth of A. thaliana: you mention that CH34 resulted in positive effects such an increase in root growth: indeed for the conditions without copper, it is visible. Also at 25 µM it is still visible, however, at 50 µm, you see a decrease compared to the conditions with 25 µM and this drop is higher than the drop that you see in similar conditions without CH34. Also, for the secondary roots: without copper, the levels with CH34 are significantly lower than without CH34. With the addition of 25 µM copper, the number does not change compared to the conditions without copper but afterwards, you a decrease. Same as for most other conditions where you already see a significant effect of CH34 in conditions without copper (and in most cases, except for root growth, the effect is negative). In conditions with 50 µM of copper, the effect clearly disappeared as the decrease from 25 µM to 50 µM is much more than that in conditions without CH34. So except for root growth, it rather has a negative impact on plant growth? Showing results with heat-killed cells would valuable for the reader.

Some smaller remarks:
Line 182: general remark but the expression of the 16S rRNA gene has been shown not be stable and expression levels can vary depending on the conditions. One of the uvrD genes are much better housekeeping genes for CH34.
Line 230-231: any idea why you don’t see an increase in secondary roots?
Line 256 -257: Cu-concentration dependent mode? You see similar effects without CH34?
Line 283: CH34 promote plant growth? Only root length?

---

## Round 0.2 · Minor Revisions

There are some additional minor comments I would like you to consider. First, can you address the issue of CH34's inability to grow on sugar (sucrose was in the growth medium) and the impact that this may have on the conclusions (rhizosphere vs. non-rhizosphere)? Second, would you consider revising the discussion to incorporate more of your results and how they relate to previous research?

·

Basic reporting

no comment

Experimental design

no comment

Validity of the findings

no comment

Additional comments

The quality of the manuscript significantly improved compared to the previous version. Although, I believe that authors could further improve the style of the manuscript and better expose their findings. My main criticism is about the Discussion section that doesn’t seem to refer enough to the authors' results but rather to the previous study. It feels that results are not sufficiently discussed in a way that can be easily interpreted. I suggest incorporating authors' observations into the text in a better, more clear way and discuss them in the context of other studies and potential implications/applications.

Line 307-309 – why condensing all the results in one sentence. It makes the sentence really loaded. Please split. This will also better highlight the results.

Fig.1 Please provide the label of the x-axis in panels B-F --> Cu concentration [µM]

Reviewer 2 ·

Basic reporting

No comment

Experimental design

No comment

Validity of the findings

No comment

Additional comments

The authors addressed (almost) all my comments. Nevertheless, the authors do not use the delta delta Ct method as indicated in their rebuttal (which would still be interesting to show, e.g. plant Cu vs plant; plant Cu vs Cu). They use delta Ct (difference in Ct values for GOI and HK). Delta delta Ct uses Ct values for GOI and HK for both the treated and control samples.

Reviewer 3 ·

Basic reporting

English grammar is much approved compared to the first version.

Experimental design

no comments

Validity of the findings

* The same medium was used in all experiments therefore gluconate was always present. This is clearly stated in lines 163-165.
However, Line 110-112: Square Petri dishes were prepared with half-strength Murashige-Skoog medium (MS1/2) (Murashige & Skoog, 1962), 0.8% agar, and 1.5% sucrose; were either inoculated or not inoculated with strain CH34.
This medium is with sucrose instead of gluconate so CH34 will not grow. And this is for most conditions throughout the paper. How can it optimally deal with the copper concentrations and influence plant growth? Results can be biased as it is able to grow on root exudates but in the non-rizosphere environment it has no carbon source to grow. This can have an important effect on the results.

Line 210: the 10,3 % is not significantly different than without CH34 inoculation so the number of rosetta’s does not increase with CH34. The fact that the difference is significant at 24 µM is because in the absence of CH34, the number of rosetta’s decreases, while with CH34, it remains the same as without Cu. That should be better explained in the text. Similarly for the secondary roots.

Additional comments

Line 33: Cop -> cop
Line 74: across
Line 75: remove over 20 different as at line 70 at least 24 metal resistance… is stated
Line 82: it is not clear what is meant by “the mobility of trace metals that are relevant to plant nutrition in this bacterial model”. Please rephrase
Line 89: A. thaliana is not in bold
Line 105: A. thaliana is not in bold
Line 277: Please change copSRABC in copD2C2B2A2R2S2 with the 2 in subscript as this is the official name and order of the cluster
Line 227: Remove “the sake of”
Line 364: may be explained by
Line 369: rephrase this sentence. Strains do not encode organic compounds. The encode genes to degrade organic compounds

---

## Round 0.3 · accepted · Accept

Thank you for your efforts to revise the manuscript in response to reviewer comments.